# Clinical Considerations for Implanted Neurological Devices in Patients Undergoing Hyperbaric Oxygen Therapy: A Case Report and Review of Manufacturer Guidelines

**DOI:** 10.3390/ijerph20095693

**Published:** 2023-05-01

**Authors:** Simone Schiavo, Connor T. A. Brenna, Anuj Bhatia, William J. Middleton, Rita Katznelson

**Affiliations:** 1Department of Anesthesiology & Pain Medicine, University of Toronto, Toronto, ON M5G 1E2, Canada; 2Hyperbaric Medicine Unit, Toronto General Hospital, Toronto, ON M5G 2C4, Canada; 3Department of Anesthesia and Pain Management, University Health Network, Toronto, ON M5G 2C4, Canada

**Keywords:** hyperbaric oxygen treatment, implanted devices, intrathecal drug delivery pumps, neurostimulators, risk assessment, case report

## Abstract

Patients with implanted medical devices are increasingly referred for hyperbaric oxygen therapy (HBOT), and the safety of exposing some of these devices to hyperbaric environments has not previously been explored. There is a paucity of evidence surrounding the management of implanted neurological devices such as neurostimulators and intrathecal drug delivery (IDD) pumps in the context of HBOT. However, these devices can be expected to harbor unique risks; for example, vacant space in the reservoir of an implanted IDD pump may change in pressure and volume during the compression and decompression phases of HBOT, resulting in a damaged or dysfunctional device. We present the case of a 27-year-old woman with cerebral palsy referred for HBOT to manage a necrotizing soft tissue infection cultured from a dehiscent abdominal wound at the previous implantation site of an intrathecal baclofen pump. An HBOT protocol was ultimately chosen in partnership with the patient and her family, but treatment was not performed due to a paucity of evidence that the implanted IDD pump could safely withstand hyperbaric exposure. In this review, we have synthesized manufacturer recommendations regarding the management of implanted neurological devices before, during, and after HBOT to inform future decision-making in this setting. Among these recommendations, we highlight that neurostimulators should be switched off for the duration of HBOT and implanted pumps should be refilled prior to each treatment session to minimize empty reservoir space.

## 1. Introduction

Implanted medical devices are increasingly prevalent, and as many as half of medical inpatients may have them [1]. The reason for this is twofold: first, with advancing technology, these devices are becoming more sophisticated, and this has led to a commensurate expansion of their applications in clinical care. Second, many devices such as surgical plates, intra-uterine devices, prosthetic joints, pacemakers, and nerve stimulators are placed with the expectation that they will remain imbedded in the body for long periods of time (years or decades). Implanted neurological devices, in particular, have garnered significant attention during the early 21st century. By modulating the activity and function of neurological structures such as the spinal cord, deep brain areas, and peripheral or cranial nerves, these devices have found unique application in the treatment of various neurological and psychiatric disorders [2,3,4,5].

Many synthetic implants are made of minimally bioreactive and corrosive-resistant metal alloys fulfilling structural or electronic functions [6]. Implantable drug delivery (IDD) pumps, which have been available in practice for several decades, differ in that they also uniquely house a fluid-filled reservoir [7]. These devices consist of an electric motor and a reservoir containing medication (e.g., local anesthetics, opioids, baclofen, ziconotide, and clonidine) that is infused into the intrathecal space through a catheter connected to the pump. The intrathecal space is usually accessed from the lumbar region and tunneled subcutaneously to the pump, which itself is usually implanted in the anterior abdominal wall. With IDD pumps, less medication is required than if it were taken orally, and fewer adverse effects are seen due to lower systemic drug concentrations. However, the reservoir of the IDD pump needs to be refilled with the medication for intrathecal infusion every few weeks by accessing a membrane in the pump reservoir via a percutaneous needle. Recently, IDD pumps have emerged alongside implantable neurostimulators as interventional therapies for chronic pain and spasticity that cannot be well managed with non-invasive pharmacological and non-pharmacological techniques [8]. Consequently, patients with these devices are presenting with increasing frequency for hyperbaric oxygen therapy (HBOT)—typically for unrelated comorbid chronic pain, although prevalence may increase dramatically as recent literature has described HBOT for chronic pain itself [9,10].

HBOT consists of breathing 100% oxygen (O_2_) in a hyperbaric chamber at pressures two to three times greater than atmospheric pressure (defined as 1 atmosphere absolute, ATA, at sea level). Central to the mechanism of HBOT are gas laws describing the relationship between the volume, pressure, and temperature of gases during therapy. For example, the increase in hydrostatic pressure within a hyperbaric chamber reduces the volume of any gas-filled spaces (under constant temperature) [11], as governed by Boyle’s Law in Equation (1):P_1_V_1_ = P_2_V_2_
(1)
where P represents pressure, V represents volume, and the subscripts _1_ and _2_ refer to unique sets of conditions a gas might be subjected to. The total pressure exerted by a mixture of gases is the sum of the pressures that would be exerted by each of the gases in isolation if they occupied the total volume, as stated by Dalton’s Law in Equation (2):P_total_ = ΣP_i_(2)
where P_total_ represents the total pressure of a gaseous mixture and P_i_ represents the partial pressures of each gas within that mixture. Thus, increasing the hydrostatic pressure within a hyperbaric chamber will elevate the partial pressures of any gases within the chamber, including O_2_. By increasing the partial pressure of O_2_ in the alveoli, HBOT results in a marked increase in the amount of dissolved O_2_ carried in the blood (rather than bound to hemoglobin). This is described by Henry’s Law in Equation (3):C = *k*P(3)
where C represents the concentration of a dissolved gas, *k* represents the gas solubility constant, and P refers to the partial pressure of a gas above a liquid. As a result, the arterial partial pressure of oxygen (PaO_2_) may be higher than 2000 mmHg during HBOT, with dissolved oxygen content increasing from 0.3 to 6.8 mL O_2_/100 mL of blood. Since the dissolved O_2_ is independent from hemoglobin, it can permeate tissues with poor or compromised circulation, significantly increasing PaO_2_ in tissues that are otherwise relatively ischemic [12].

The conditions currently approved for treatment with HBOT by both the Undersea and Hyperbaric Medicine Society and Health Canada are summarized in Table 1 [13].

In addition to these approved indications, HBOT has shown benefits as adjunctive therapy for selected complex neurological and chronic pain conditions because of its multiple primary and secondary beneficial effects (Figure 1) [10]. 

HBOT treatments are usually administered daily for 90 to 120 min, over 20 to 40 consecutive sessions. Overall, HBOT is considered a safe, non-invasive therapy with very few adverse effects (Table 1) [14]. The major absolute contraindication to HBOT is an untreated pneumothorax because, due to the gas laws described above, this condition can progress with changes in environmental pressure and has the potential to develop into a tension pneumothorax.

Implanted neurological devices, especially those containing reservoirs (e.g., IDD pumps), may be expected to undergo similar changes in pressure and volume during HBOT. However, reports of patients undergoing HBOT with implanted neurological devices are sparse [15,16,17,18], and this consideration remains largely uninvestigated. We present here a case of a patient with an intrathecal baclofen pump who was referred to our unit for an abdominal wound infection and an infected pump implantation site. We also provide recommendations regarding the use of HBOT in patients with implanted IDD pumps and neurostimulators.

## 2. Detailed Case Description

A 27-year-old woman with cerebral palsy and severe spasticity presented to the Emergency Department for abdominal wound dehiscence. Her baseline neurological status was a Glasgow Coma Scale of 8 (E4V1M3). Her past medical history was significant for absence seizures with apnea, multiple previous thoracolumbar spine surgeries, bilateral hip surgeries, a stage IV sacral pressure ulcer, a percutaneous feeding gastrostomy tube in situ, and a past episode of aspiration pneumonia complicated by septic shock requiring prolonged ICU admission. She presented with an intrathecal baclofen pump (SynchroMed II 40 mL, Medtronic PLC, Minneapolis, MN, USA), initially implanted 13 years prior to treat her spasticity and subsequently replaced twice: six years and one year prior to her emergent presentation. After her most recent pump replacement, the patient developed a series of infectious complications, requiring multiple surgeries, including abdominal washout and exposure of the pump embedded in a lower abdominal quadrant.

In the Emergency Department, a draining abdominal wound at the site of pump insertion was identified, and the patient was brought to the operating room for further irrigation and debridement. Intraoperatively, the surgical team noted evidence of frank wound dehiscence and erosion of the skin overlying the pump. The pump was removed, and a similar, smaller version (SynchroMed II 20 mL, Medtronic PLC, Minneapolis, MN, USA) was inserted in its place. Wound swabs were collected, and a referral was made to the Hyperbaric Medicine Unit (HMU) at the Toronto General Hospital, University Health Network, Toronto, Canada, for treatment of a provisional necrotizing soft tissue infection. Later, the swabs were reported as positive for *Corynebacterium* species, and targeted antibiotic therapy was initiated.

HBOT was indicated for this patient on an urgent basis to treat progressive bacterial gangrene in the context of a possibly infected implanted device. Given this patient’s heightened risks of respiratory complications and oxygen toxicity, treatment was considered with a modified *Table 9 US Navy* (the standard protocol used for the majority of HBOT indications) [13]. This protocol consists of breathing 100% O_2_ at a pressure of 2.4 ATA for 90 min, interrupted by two 5 min “air breaks” to decrease the risk of hyperoxic complications. The treatment team proposed a modification of three “air breaks” with the aim of further reducing seizure risk. Alternative protocols such as shorter, 60 min sessions or decreased pressures such as 1.8–2.0 ATA were also discussed as possible avenues of risk mitigation. Additionally, given the patient’s inability to perform equalization maneuvers, the hospital’s otolaryngology service was consulted and made themselves available to perform elective bilateral myringotomies on the first day of treatment to preclude ear barotrauma.

Theoretical risks posed by HBOT to the implanted IDD pump were considered, including a concern that air cavities within the device could change in size or shape during compression and decompression. Unfortunately, in the absence of evidence that such a device could safely undergo hyperbaric exposure, the hyperbaric medicine unit safety director elected not to proceed with HBOT. The patient was repatriated to the referring hospital and discharged soon afterward. Two months later, the patient was emergently admitted with an abdominal blister rupture and re-erosion of the pump through the abdominal fossa. Her family reported a persistent dark discoloration of her abdomen with blisters overlying the pump site for several weeks prior to readmission. During this final admission, the patient was gradually weaned off of intrathecal baclofen and transitioned to antispasmodic therapy (baclofen and dantrolene) administered via G-tube. The pump was eventually removed, uneventfully and with subsequent abdominal wound healing.

## 3. Discussion

HBOT may be considered a therapy for several Health Canada-approved indications in patients who incidentally also have an implanted neurological device, like this case of an infection in a patient with an implanted IDD pump. Patients with an indwelling IDD pump or a spinal cord stimulator may also be considered for HBOT as an adjunctive therapy for the same condition requiring the device, such as complex regional pain syndrome. Some device manufacturers provide pertinent information regarding the function of their devices under hyperbaric conditions, typically geared towards scuba diving. Nonetheless, a thorough individual assessment of risks and benefits must be performed based on the specific device, the strength of the clinical indication for HBOT, local training and safety protocols, as well as any protocols recommended in the literature [19]. At our center, the typical process followed in the case of implanted devices includes checking the device model and technical specifications and receiving clearance from three parties: the device manufacturer (if possible), the hospital biomedical engineering department, and the hyperbaric medicine unit safety director. If concerns are raised by any party that cannot be resolved, as happened in this case, HBOT is not performed.

Neurological devices include a broad range of neurostimulation devices implanted in the brain, spinal cord, and peripheral or cranial nerves to modulate neural activity, as well as a variety of infusion pumps, which are typically used for the intrathecal continuous administration of drugs to treat chronic pain or spasticity. Depending on the specific therapeutic objectives, a variety of devices are available for implant. For example, to treat severe neuropathic chronic pain, invasive therapeutic options include spinal cord stimulators (SCS) [20], dorsal root ganglion stimulation (DRGS), sacral or peripheral nerve stimulators (SNS/PNS) [2], and intrathecal drug delivery (IDD) pumps [5]. Overall, regardless of the specific indication, the use of these devices is increasing exponentially [21], and the frequency of patients with implanted stimulators or pumps presenting for adjunctive HBOT can be expected to increase in parallel [15,16]. This trend mirrors the well-known and dramatic increase in HBOT referrals for patients with implanted Permanent Pacemakers or Implantable Cardioverter Defibrillators over the last few decades [22,23]. Initially, concerns relating to a plausible fire risk of devices emitting electrical impulses within a hyperbaric chamber (containing pure O_2_) were highlighted for these devices [19,24]. However, several reports and manufacturers’ technical data have since confirmed the safety of these devices in the hyperbaric setting [16,24,25,26]. Additional risks of malfunction related to structural damage or performance deterioration due to the changes in pressure have also been highlighted [19,23].

Implanted pumps such as IDD pumps are unique even among these neurological devices, with an additional consideration arising from the possibility that hyperbaric exposure could result in a change in reservoir size during compression and decompression phases of treatment, leading to rupture or warping of the device and subsequent malfunction or improper refilling and storage of medication for subsequent use [15,17,18,27,28]. From a technical point of view, prior to the application of HBOT, formal clearance should be obtained from the manufacturer whenever possible regarding feasibility and maximum tolerable pressure for each specific device to allow for accurate assessment and minimization of possible complications such as hardware malfunctions or reservoir deformities.

While there is no current evidence contraindicating HBOT for patients with neurostimulation devices and pumps, the feasibility and tolerability of treatment are based on the specific model of the device in place, with the potential to require limitations in the pressure (typically to a maximum of 2.0 ATA) and length of HBOT, as well as the usual close monitoring of the patient undergoing treatment. To clarify and summarize different devices’ limitations with respect to hyperbaric exposure, we surveyed eight major neuro-implantable device suppliers whose devices are used around the world regarding unique considerations for HBOT. Detailed specifications and recommendations from these manufacturers for the management of implantable pain devices prior to and during HBOT are summarized in Table 2. These recommendations represent a summary of our survey and should not be used in isolation to guide clinical decision-making without the formal involvement of a device’s manufacturer.

In addition to device-specific abilities to withstand hyperbaric pressure, general recommendations were to switch neurostimulation devices off for the duration of HBOT sessions and to refill IDD pumps prior to initiating HBOT. The present report describes the case of a patient with an IDD pump, for which the specific recommendation provided by the manufacturer [30] for future cases was to have a specialist nursing team perform daily interrogations of the device and document its administration rate and residual volume. They further advised:

“*It is possible to damage the pump due to deformation of the bottom shield of the pump if it is exposed to pressures greater than 2.0 ATA for a single treatment, or at pressures less than or equal to 2 ATA with multiple treatments if the reservoir is not kept close to full; a full reservoir helps to support the bottom shield of the pump. [While the functioning of the pump is not affected by HBOT] a collapsed bottom shield may significantly reduce the pump’s drug capacity and would require pump replacement surgery. [Therefore, the suggested management of the pump would be to] keep the reservoir volume within 2 mL of the recommended [maximum] capacity of the [20 mL Medtronic SynchroMed] pump, and it should be refilled during the hyperbaric treatment period to keep it at or above these volumes*”.

## 4. Conclusions

When a patient with an implanted neurological device is referred for HBOT, the device’s manufacturer should be contacted to provide formal clearance regarding its feasibility, safety, and best practices for management in the peri-HBOT period. There is currently a paucity of available guidelines and recommendations, but we identify several key concepts common to the majority of neurological devices, including the deactivation of neurostimulation devices and filling of implanted pumps prior to starting HBOT and the fastidious monitoring of patients with implanted devices in the pre-, intra-, and post-HBOT periods. IDD pumps, in particular, warrant cautious treatment decisions given the presence of air cavities within these devices. Wherever possible, the involvement of the manufacturing company’s representatives as well as close communication with them prior to and following each session is a valuable source of support, while the possible nuances of HBOT for patients with devices remain largely unknown. Daily interrogations of the pump or the stimulator by clinical personnel may also be helpful. Finally, in addition to risk-benefit considerations made in partnership with patients and in view of the device manufacturer’s recommendations, safety clearance should be obtained per local policies (for example, by involving institutional safety directors in the planning and execution of treatment for these patients).

## Figures and Tables

**Figure 1 ijerph-20-05693-f001:**
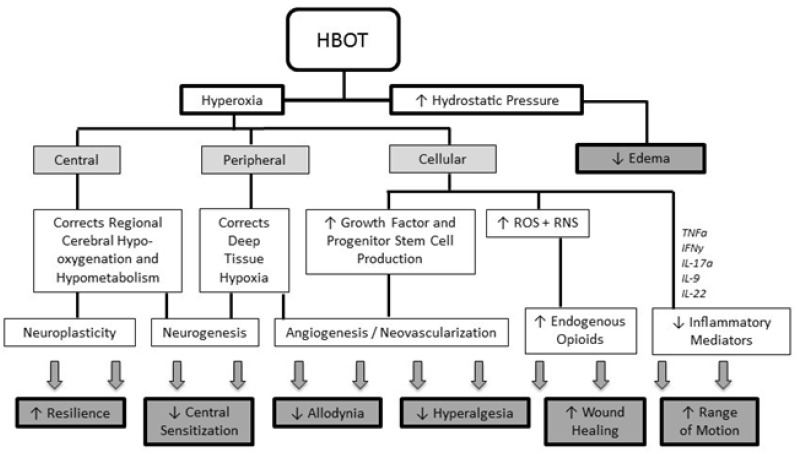
Overview of the multiple primary and secondary effects of HBOT (reproduced with permission from Schiavo et al., 2021 [10]). HBOT: Hyperbaric Oxygen Therapy; ROS: reactive oxygen species; RNS: reactive nitrogen species; TNFα: tumor necrosis factor—alpha; IFNγ: interferon-gamma; IL: interleukin.

**Table 1 ijerph-20-05693-t001:** Conditions currently approved for treatment with HBOT by the Undersea and Hyperbaric Medical Society [13] and by Health Canada; side effects of HBOT [14].

Indications for Hyperbaric Oxygen Therapy
Air or gas embolisms
Carbon monoxide poisoning
Clostridial Myositis and Myonecrosis
Crush injury, Compartment Syndrome and other acute traumatic ischemia
Decompression sickness
Arterial insufficiencies (Central Retinal Artery Occlusion, Enhancement of healing in selected problem wounds)
Severe Anemia
Intracranial abscess
Necrotizing soft tissue infections
Refractory osteomyelitis
Delayed radiation injury (Soft Tissue and Bony Necrosis)
Acute Thermal Burn Injury
Idiopathic Sudden Sensorineural Hearing Loss
**Side Effects of Hyperbaric Oxygen Therapy**
Middle ear barotrauma, Sinus squeeze, Claustrophobia, Progressive myopia, Pulmonary barotrauma, Seizures

**Table 2 ijerph-20-05693-t002:** Key concepts and specifications for implanted neurostimulation systems and IDD pumps in HBOT.

	Check HBOT feasibility with manufacturer (see below). Turn the stimulator OFF while receiving HBOT (* only Medtronic devices could be left ON). Refill the IDD pump reservoir before HBOT.		
**Manufacturer**	**Device (Models)**	**HBOT Feasibility and Maximum ATA Exposure**	**Notes**
Abbott (ex St. Jude Medical)	**SCS**: All models (Proclaim, Prodigy, Eon, EonC, EonMini, Protégé, Genesis, GenesisXP, GenesisRC)	4.0 ATA [29]	Turn devices OFF during HBOT
	**DBS**: All models (Infinity, Libra, LibraXP, Brio)		
	**SCS-DRG**: All models (Proclaim DRG, Axium DRG)	1.5 ATA [29]	
Medtronic	**Stimulators**: All models including SCS (Intellis, PrimeAdvance), SNS (InterStim), PNS, DBS (Activa, Restore),	2.0 ATA [30]	These devices can be left ON during HBOT
	**IDD pump** (SynchroMed II)		IDD pump must be refilled during HBOT in case of multiple sessions
Boston Scientific	All models	No recommendations	No tests performed in humans, device functioning not guaranteed [31,32,33]
Stimwave technologies	**SCS** (Freedom)	1.5 ATA [34]	*Instructions for use* recommends 1.5 ATA, but also 13 m (45 feet) for diving, which would correspond to 2.3 ATA; no clarifications received by manufacturer [35]
Flowonix	**IDD pump** (Prometra II)	2.0 ATA [36]	*Instructions for use* states “an increase in environmental pressure of approximately 1 ATA or greater may cause the pump to temporarily stop delivering drug; when normal atmospheric pressure is returned, the pump will resume its programmed delivery rate” [37]
Nevro	**SCS**: All models (Senza, Senza II, Senza Omnia)	4.5 ATA [38]	Turn devices OFF during HBOT [39]
Saluda Medical	**SCS** (Evoke)	1.5 ATA [40]	No response from vendor
Nalu	**SCS/PNS** (Nalu)	1.48 ATA [41]	*Instructions for use* recommends 1.48 ATA, but also 13 m (45 feet) for diving, which would correspond to 2.3 ATA; no clarifications received by manufacturer Do not dive/hyperbaric the external Therapy Disc

IDD: intrathecal drug delivery; SCS: spinal cord stimulator; DBS: deep brain stimulator; SNS: sacral nerve stimulator; PNS: peripheral nerve stimulator; DRG: dorsal root ganglion.

## Data Availability

Data can be available upon reasonable request to the corresponding author.

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
