# Peer review of "Clinical Considerations for Implanted Neurological Devices in Patients Undergoing Hyperbaric Oxygen Therapy: A Case Report and Review of Manufacturer Guidelines"

_ijerph, 2023, doi:10.3390/ijerph20095693_

Round 1
Reviewer 1 Report
The manuscript by Schiavo et al. addresses an interesting subject for which there is a lack of comprehensive information. Subjects with implanted devices are appearing more frequently at hyperbaric centres and each needs to be considered individually taking account of manufacturer safety information where available. The manuscript describes a case presentation of a patient with an implanted device who was subsequently not treated and goes on to describe some information gained prom manufacturers regarding the suitability of their devices for use in a hyperbaric environment. The subject matter merits publication but there are elements of the manuscript that would benefit from restructuring.
1. Introduction
L106: … ‘reports of patients undergoing HBOT with medical devices are sparce [12]’.
In addition to reference 12 which discusses hardware related infections there are numerous other publications on the subject, many of which are referenced in this manuscript (references 17-25) and some of which report the issues with intrathecal pumps under hyperbaric conditions. Consider expanding the references for this statement with some description of some of the issues encountered. Consider moving some of the descriptive section in the discussion (referring to refs 17-25) into the introduction at this point.
2. Case Description
L129: ….. SynchroMed II 20 mL, Medtronic PLC …..
L150-151: ….absence of evidence that such a device could safely undergo hyperbaric exposure …….
Table 2: SynchroMed pump maximum 2 ATA and should be refilled during HBOT
The case description focusses on a patient with an implanted Synchromed II pump which is suitable for compression to 2 ATA but treatment was not conducted due to lack of evidence that the pump could safely undergo hyperbaric exposure. As presented here these statements appear contradictory and should be explained.
3. Discussion
L211: Table 2 ….. implanted neurostimulation systems and ITP in HBOT.
The term ITP has not been used previously and potentially causes confusion. Suggest change to IDD pumps as used elsewhere. Also applies to L215.
Table 2 content and further information
The authors should reconsider how they represent the information presented in Table 2. Using the Medtronic SynchroMed II as an example (SynchroMed II noy Synchromed II?), it is clearly stated that the pump has a maximum exposure of 2ATA and must be refilled during HBOT. This is potentially misleading when taken against the text supplied by the manufacturer which provides more detail on multiple exposures. Several of the specifications in Table 2 are based on personal communications by email that cannot be properly referenced or read by anybody considering the safety of exposing patients with such devices to hyperbaric therapies. It is my opinion that there should be a clear statement addressing this and that none of the specifications in the table should be used without contacting the manufacturer for further details. Table 2 is useful as it adds substance to the issue being addressed but it should not be used as reference material in the treatment of patients. The involvement of manufacturers is appropriately addressed in the discussion but should be emphasised in the legend to the table itself.
Two minor comments –
L91-92: Figure 1 is reproduced with permission but from where?
L102: Medical Society (not Medicine Society). Also reference 10 relates to the UHMS indications and not those of Health Canada (although they may adopt the same). Suggest moving [10] to immediately after Society.
Author Response
Reviewer 1
Comments and Suggestions for Authors
The manuscript by Schiavo et al. addresses an interesting subject for which there is a lack of comprehensive information. Subjects with implanted devices are appearing more frequently at hyperbaric centres and each needs to be considered individually taking account of manufacturer safety information where available. The manuscript describes a case presentation of a patient with an implanted device who was subsequently not treated and goes on to describe some information gained prom manufacturers regarding the suitability of their devices for use in a hyperbaric environment. The subject matter merits publication but there are elements of the manuscript that would benefit from restructuring.
- Introduction
L106: … ‘reports of patients undergoing HBOT with medical devices are sparce [12]’.
In addition to reference 12 which discusses hardware related infections there are numerous other publications on the subject, many of which are referenced in this manuscript (references 17-25) and some of which report the issues with intrathecal pumps under hyperbaric conditions. Consider expanding the references for this statement with some description of some of the issues encountered. Consider moving some of the descriptive section in the discussion (referring to refs 17-25) into the introduction at this point.
Answer: Thank you for the comment. We have revised this sentence for added clarity, in the revision, and have expanded the references accordingly to present the existing literature on neurological devices (L 113). However, we decided that our manuscript’s discussion of the mechanisms and effects of pressure on implanted devices should remain in the Discussion (L 198-203), as it provides details and explanations which we feel would make the Introduction too long and difficult to read without the context provided by our clinical case in the center of the submission.
- Case Description
L129: ….. SynchroMed II 20 mL, Medtronic PLC …..
Answer: Manufacturer details were added (L 125)
L150-151: ….absence of evidence that such a device could safely undergo hyperbaric exposure …….
Table 2: SynchroMed pump maximum 2 ATA and should be refilled during HBOT
The case description focusses on a patient with an implanted Synchromed II pump which is suitable for compression to 2 ATA but treatment was not conducted due to lack of evidence that the pump could safely undergo hyperbaric exposure. As presented here these statements appear contradictory and should be explained.
Answer: Thank you for identifying that this section can benefit from added clarification. In our presented case, the decision “not to treat” was made by the Hyperbaric Unit safety director due to a lack of available evidence for the feasibility of the process suggested by the manufacturer (daily interrogation and possible refill), which had not been reported in the literature. This is among the novel contributions of our submitted manuscript, which aims to present the clinical considerations surrounding implanted neurological devices and HBOT, as well as guidelines from common device manufacturers, so that providers faced with similar decisions in the future will have an actionable resource to guide decision-making. For added clarity to this, we have modified the sentence accordingly (L 158-159). The process followed at our institution in case of patients with implanted devices dictates a triple approval; one of them comes from the hyperbaric unit safety director, as better explained in Discussion (L 179-184).
- Discussion
L211: Table 2 ….. implanted neurostimulation systems and ITP in HBOT.
The term ITP has not been used previously and potentially causes confusion. Suggest change to IDD pumps as used elsewhere. Also applies to L215.
Answer: Thank you for your attention to detail: this was a typographical error in our original submission. “ITP” has now been changed to “IDD pump”, and IDD is an abbreviation already introduced and used in the manuscript.
Table 2 content and further information
The authors should reconsider how they represent the information presented in Table 2. Using the Medtronic SynchroMed II as an example (SynchroMed II noy Synchromed II?), it is clearly stated that the pump has a maximum exposure of 2ATA and must be refilled during HBOT. This is potentially misleading when taken against the text supplied by the manufacturer which provides more detail on multiple exposures. Several of the specifications in Table 2 are based on personal communications by email that cannot be properly referenced or read by anybody considering the safety of exposing patients with such devices to hyperbaric therapies. It is my opinion that there should be a clear statement addressing this and that none of the specifications in the table should be used without contacting the manufacturer for further details. Table 2 is useful as it adds substance to the issue being addressed but it should not be used as reference material in the treatment of patients. The involvement of manufacturers is appropriately addressed in the discussion but should be emphasised in the legend to the table itself.
Answer: Thanks for pointing this out. We added the words “in case of multiple sessions” in Table 2 (for Medtronic - IDD pump – Notes). Also, we want to reassure the reviewer that the concept of using/not using the paper’s specifications appropriately, was explicitly clarified by stating that the Table is a summary of different devices’ limitations as per a survey of the major manufacturers (L 218-221); we also clearly stated in Table 2, as the first key concept: “check HBOT feasibility with manufacturer” (L 229). However, based on the reviewer suggestion, our revised manuscript includes an additional proviso to explicitly highlight that none of the specifications in the table should be used without contacting the manufacturer for further details, close to Table 2 (L 223-225).
Two minor comments –
L91-92: Figure 1 is reproduced with permission but from where?
Answer: The original figure’s source is referenced in the title of the Figure itself (now #Ref 10); to add clarity, the author has been explicitly mentioned in the Figure Legend, and the reference has been moved to appear after the words “reproduced with permission” rather than before these words. Also, a specific Copyright-Reproduction Permission form was originally submitted to the journal of original publication, as requested; see below a screenshot of the original Permission Form. We confirm that authors retain full copyright of articles published in this journal, and have permission to reproduce the original work with attribution as we have done in this case.
L102: Medical Society (not Medicine Society). Also reference 10 relates to the UHMS indications and not those of Health Canada (although they may adopt the same). Suggest moving [10] to immediately after Society.
Answer: Thank you, we have corrected the misspelling and have moved the reference as suggested (L 93).
Reviewer 2 Report
The authors present a study on the effects of Hyperbaric Oxygen Therapy (HBOT) in patients that have implanted neurological devices, and with particular emphasis in intrathecal drug delivery (IDD) pumps. They highlight the case of a patient who required HBOT treatment but was deemed dangerous due to the presence of an IDD pump.
The article highlights a potential danger in using HBOT treatment in such implants, and their point is explained clearly. Furthermore, it is presented in a useful way, with proposed future steps to take in similar cases.
This topic has been covered by Kot, Jacek. "Operation of Implantable Cardiac Devices in Hyperbaric Conditions." Pacemakers and Defibrillators. IntechOpen, 2022. Although this reference focused on cardiac implants in HBOT treatment, it also provides general steps to approach the use of HBOT treatment with any implantable device.
In addition, risks of using implantable devices under hyperbaric conditions have been already highlighted and recommendations of use explained in European Regulations (Eurpean Norm CEN EN14931). Kot listed the potential hazards that may occur when a implanted device exist during HBOT treatment.
This article could be strengthen by referring to the reference provided and highlighting the main hazards and steps listed for safety of use of HBOT in such devices, and add their own findings regarding the use of IDD pumps, which is the added information provided.
Other references that coincide with the aforementioned review are:
Gawdi, Rohin, and Jeffrey S. Cooper. "Hyperbaric contraindications." StatPearls [Internet]. StatPearls Publishing, 2022.
Bartek Jr, Jiri, et al. "Hyperbaric oxygen therapy as adjuvant treatment for hardware-related infections in neuromodulation." Stereotactic and Functional Neurosurgery 96 (2018): 100-107.
Few other minor corrections are attached.

Author Response
Reviewer 2
Comments and Suggestions for Authors
The authors present a study on the effects of Hyperbaric Oxygen Therapy (HBOT) in patients that have implanted neurological devices, and with particular emphasis in intrathecal drug delivery (IDD) pumps. They highlight the case of a patient who required HBOT treatment but was deemed dangerous due to the presence of an IDD pump.
The article highlights a potential danger in using HBOT treatment in such implants, and their point is explained clearly. Furthermore, it is presented in a useful way, with proposed future steps to take in similar cases.
This topic has been covered by Kot, Jacek. "Operation of Implantable Cardiac Devices in Hyperbaric Conditions." Pacemakers and Defibrillators. IntechOpen, 2022. Although this reference focused on cardiac implants in HBOT treatment, it also provides general steps to approach the use of HBOT treatment with any implantable device.
In addition, risks of using implantable devices under hyperbaric conditions have been already highlighted and recommendations of use explained in European Regulations (Eurpean Norm CEN EN14931). Kot listed the potential hazards that may occur when a implanted device exist during HBOT treatment.
This article could be strengthen by referring to the reference provided and highlighting the main hazards and steps listed for safety of use of HBOT in such devices, and add their own findings regarding the use of IDD pumps, which is the added information provided.
Answer: We thank the reviewer for their interest in our manuscript, and for their positive comments. We agree that some of the discussion points have been previously touched upon by Kot’s book chapter, which focuses on cardiac devices (rather than neurological devices and implanted pumps) and relative risks. In our revised manuscript, we have added this book chapter among our references (now ref #19).
Although many issues related to cardiac devices are similar and common to any other implanted device, the details discussed in Kot’s chapter are very specific to pacemakers and ICDs (e.g., risk of fire by the electric arc initiated by the ICD) and not completely superimposable to the neuro devices. The European laws reported in this chapter have a limited applicability to North American practice, although appropriate for these devices. Also, it is reassuring that the majority of references in Kot’s chapter were already cited (and still are) in our paper, including Kot’s previous articles on the same topic from 2005 and 2014; as suggested, we have nevertheless added several more references relevant to neuro devices and to our North American context, such as refs #23 (Lafay) and 26 (Schmitz). Overall, taking into consideration the suggestions and the reference provided by the reviewer, we have adjusted and expanded the main hazards and steps listed for safety of HBOT use in patients with these devices, presented in our manuscript’s Discussion (L 178 and 201-203).
Other references that coincide with the aforementioned review are:
Gawdi, Rohin, and Jeffrey S. Cooper. "Hyperbaric contraindications." StatPearls [Internet]. StatPearls Publishing, 2022.
Answer: The suggested reference discusses in general the contraindications to HBOT, and we have a similar and more appropriate (book chapter) reference to support this information (ref #14 Jain, K. K., Indications, Contraindications, and Complications of HBO Therapy. In Textbook of Hyperbaric Medicine, Jain, K. K., Ed. Springer: 2017; pp 79-84.). Also, the suggested reference does not contain any information related to implanted devices, on which our manuscript focuses, and does not cite any literature relevant to this topic. On the contrary, our present work collates relevant evidence pertaining to this subject which spans widely from cardiac devices (Kot J, ref #19, 24), to general principles (Kot J, ref #16), to specific neuro devices (Bartek, ref #15).
Bartek Jr, Jiri, et al. "Hyperbaric oxygen therapy as adjuvant treatment for hardware-related infections in neuromodulation." Stereotactic and Functional Neurosurgery 96 (2018): 100-107.
Answer: Thank you for your comment. This reference was already cited and present among our references (previously ref #12, now #15).
Few other minor corrections are:
L 42: It would be interesting to see additional references for other neurological stimulation devices, as maybe their real application is more common than deep brain implants
Answer: Thank you for the suggestion. The original reference to a book chapter was inadvertently and mistakenly replaced with a single article, and we are grateful to the reviewer for identifying this error. We have fixed the references by adding the appropriate book chapter (#2), and by expanding the bibliography with the inclusion of additional relevant literature on SCS, DBS and IDD pumps (ref #3-5).
L43-49:
While many synthetic implants are made of minimally-bioreactive and corrosive-resistant metal alloys fulfilling structural or electronic functions [3], implantable drug delivery pumps (IDD) - which uniquely house a fluid reservoir - have been available in practice for several decades [4]. The IDD pump consists of an electric motor and a reservoir containing the medication (e.g., local anesthetics, opioids, baclofen, ziconotide, clonidine) that is infused in the intrathecal space through a catheter connected to the pump.
This phrase doesn't seem to hold a good relation between its comparative terms (the use of corrosive resistant materials vs IDD that house a fluid reservoir) and therefore the meaning is not very clear
Answer: We have re-stated the sentence to add clarity (L 45-49)
L78 solubility constant
Answer: We changed the phrase “k represents Henry’s law constant”, to “k represents the gas solubility constant”.
L101 Table 1 should appear before figure 1
Answer: Thank you for catching this layout/editorial issue. We have reorganized our exhibits in the revised manuscript to present Table 1 prior to Figure 1.
L 118 She presented with an intrathecal baclofen pump, [Add here the reference of pump manufacturer] initially implanted 13 years prior to treat her spasticity and replaced twice - six years and one year prior to her emergent presentation.
Answer: Thank you for catching the missing information in this passage. We have moved the reference/details about the pump specifics and manufacturer from line 134 to line 125.
L 215 ITP
Answer: Thank you for your attention to detail: this was a typographical error in our original submission. “ITP” has now been changed to “IDD pump”, and IDD is an abbreviation already introduced and used in the manuscript.